# Prurigo Nodularis: Pathogenesis and the Horizon of Potential Therapeutics

**DOI:** 10.3390/ijms25105164

**Published:** 2024-05-09

**Authors:** Hwa Jung Yook, Ji Hyun Lee

**Affiliations:** Department of Dermatology, Seoul St. Mary’s Hospital, College of Medicine, The Catholic University of Korea, #222 Banpo-daero, Seocho-gu, Seoul 06591, Republic of Korea

**Keywords:** prurigo nodularis, chronic prurigo, treatment

## Abstract

Chronic pruritus that lasts for over 6 weeks can present in various forms, like papules, nodules, and plaque types, with prurigo nodularis (PN) being the most prevalent. The pathogenesis of PN involves the dysregulation of immune cell–neural circuits and is associated with peripheral neuropathies, possibly due to chronic scratching. PN is a persistent and challenging condition, involving complex interactions among the skin, immune system, and nervous system. Lesional skin in PN exhibits the infiltration of diverse immune cells like T cells, eosinophils, macrophages, and mast cells, leading to the release of inflammatory cytokines and itch-inducing substances. Activated sensory nerve fibers aggravate pruritus by releasing neurotransmitters, perpetuating a vicious cycle of itching and scratching. Traditional treatments often fail, but recent advancements in understanding the inflammatory and itch transmission mechanisms of PN have paved the way for innovative therapeutic approaches, which are explored in this review.

## 1. Introduction

Pruritus, commonly known as itching, is characterized as an unpleasant sensation prompting a desire to scratch; it is considered as chronic when persisting beyond a period of 6 weeks [1]. Chronic pruritus undergoes its own progression and involves peripheral and central sensitization to itch, irrespective of its initial cause [1,2]. Chronic prurigo manifests in various forms, such as papules, nodules, patches, umbilical shapes, or linear forms, and these types can coexist or appear successively. The distinctions include the papular type, with pruritic lesions smaller than 1 cm in diameter; the nodular type, represented by prurigo nodularis (PN) and characterized by larger pruritic dome-shaped nodules exceeding 1 cm in diameter; the plaque type, involving pruritic flat plaques typically exceeding 1 cm and often found on the lower leg; the umbilicated type, featuring ulcers and pruritic borders; and the linear type, with lesions arranged in a linear pattern [3]. Among these, nodular prurigo is the most prevalent and typically represents the final stage. Despite their varied presentations, these clinical phenotypes of chronic pruritus share common disease characteristics, categorizing them under the overarching term of chronic prurigo [4]. In Europe, the nodular form of this disease is described as chronic prurigo of the nodular type [3]. In the United States, it is often called PN [5], comprising 70% of all chronic prurigo cases [6]. A consensus among US experts favors the inclusive term “PN” to encompass all its variants, deeming it adequately descriptive [5]. Ultimately, European and US experts collaborated to formulate the International Forum on the Study of Itch (IFSI) guideline on chronic prurigo, which includes PN. They concluded that the nodular type of chronic prurigo, PN, and chronic nodular prurigo are synonymous [7]. Japanese definitions align closely with these descriptions, albeit with slight variations [8].

Apart from causing a considerable difficulty in an individual’s quality of life, a considerable number of PN patients exhibit resistance to treatment [9]. The development of PN is believed to result from a skin reaction triggered by persistent itching and subsequent repetitive scratching [10,11]. The precise mechanisms underlying the pathogenesis of PN are not fully understood. Still, existing research indicates that a substantial interplay and dysregulation between immune cells and neural circuits are key factors in the process [12]. PN is currently diagnosed clinically, but the histopathology of nodules often reveals thickened, hyperplastic dermal nerve fibers with a decreased intraepidermal nerve fiber density. Furthermore, PN has been linked to peripheral neuropathies in epidemiologic studies. However, recent research suggests that structural changes in the epidermis may result from mechanical damage caused by chronic excoriation [9]. This review article discusses the difficulties associated with the treatment of PN and outlines newly developed pharmacological agents informed by the understanding of its pathogenesis. We aimed to present an evidence-based treatment strategy focusing on advancing personalized medicine for PN.

## 2. Epidemiology

PN is a relatively uncommon condition, with an estimated occurrence of 72 cases per 100,000 individuals in a survey of US adults aged 18 to 64 with health insurance [13]. Due to potential variations in the coding of PN in medical claims, this statistic could be a conservative approximation. Recent research indicates that PN impacts an estimated 36.7 to 148.3 individuals per 100,000 in the US [14,15], and the highest prevalence is observed among patients of color [16,17,18]. In European countries, its prevalence varies, ranging from 8 to 200 cases per 100,000 people [1,16], and the annual incidence in Germany is reportedly 0.02% [19]. In Asia, the annual incidence of PN in Korea is estimated to be 0.036% [20] and 4.82 cases per 1000 dermatology outpatients [21]. Nationwide cross-sectional and seasonal multi-center studies in Japan reported 1229 prurigo patients (1.82%) out of 67,448 total dermatological patients [8]. In an epidemiological study in the US, the average and median ages were determined to be 50.9 and 54 years, respectively [13]. Another investigation at a US institution revealed that, out of 909 PN patients, the majority fell within the 51 to 65 age range [16]. An analysis of the 2016 National Inpatient Sample dataset indicated an average age of 55.2 years for hospitalized PN patients [22]. According to real-world data obtained from Korea, individuals with PN had an average age of 58.1 years, consistent with the results of earlier studies in western populations (50.9 years in the United States [13] and 61.5 years in Poland [23]). While PN typically affects individuals in their fifth and sixth decades, there have been reported cases in pediatric patients and older adults based on small case series [24,25].

PN was initially described in a group consisting solely of female patients in the early 20th century [26]. Despite subsequent clinical observations confirming that PN affects both males and females, there is some indication that it may be slightly more prevalent in females. In a US epidemiological study involving 7095 adult PN patients, 53.1% were female, while 46.9% were male [13]. Meanwhile, variations in the gender distribution of PN based on race have been suggested. A study conducted at a single center revealed that 54.6% of African American PN patients were female, compared to 50.5% for whites and 41.9% for Asians with PN [16]. In a Korean epidemiological study, males showed an increased risk of incident PN compared with females [20], and a real-world study of PN in Korea proved male predominance (56.8%) [21].

Both research conducted at a single institution and national databases indicate a higher prevalence of PN in individuals of color [16,22]. Boozalis et al. found 3.4-fold increased odds (95% confidence interval [CI] 2.9–3.9, *p* < 0.001) of PN in African Americans compared to whites in both outpatient and inpatient settings [16]. An analysis of 2016 National Inpatient Sample data also revealed elevated odds of hospitalization for PN in black individuals (odds ratio [OR] 4.43, *p* < 0.001), Asian individuals (OR 3.44, *p* = 0.003), and Hispanic individuals (OR 1.77, *p* = 0.02) when compared to whites [22]. Additionally, African American patients with atopic dermatitis (AD), aside from being at a higher risk for PN, may exhibit a more significant number of PN lesions compared to other racial groups [27]. Suboptimal care for AD in African American patients in spite of their likelihood to experience a more persistent and severe disease course might contribute to a higher prevalence of moderate-to-severe AD, subsequently increasing the risk of concurrent PN in this population [28,29]. However, this trend is unrelated to Fitzpatrick phototypes or skin pigmentation [30]. A recent study using GWAS revealed that Black individuals face a genetic risk over two times higher for developing PN (OR 2.63, *p* < 0.0001). Notably, the association with race remained more significant than when adjusting for genetic ancestry. As race is a sociocultural construct rather than a strictly genetic category, it suggests that genetics, environmental factors, and social determinants of health likely play roles in PN development and may contribute to observed racial disparities in clinical outcomes [31]. Recognizing the disproportionate burden of PN in individuals with specific races could aid in addressing health outcome disparities for these patients.

## 3. Etiology and Pathogenesis

### 3.1. Immune Dysregulation

The pathology reveals concentrated infiltrations within the dermis of PN lesions. These infiltrates predominantly comprise elevated quantities of T lymphocytes, mast cells, and eosinophilic granulocytes [32,33]. In the skin, immune cells trigger a robust inflammatory reaction and severe itching through the release of mediators like interleukin (IL)-31, histamine, prostaglandins, eosinophil cationic protein (ECP), tryptase, and neuropeptides [12,32,33]. The immune response is pivotal in the development of PN (Figure 1). Eosinophils contribute to the cutaneous inflammation and itching linked with PN, accumulating these cells in the dermis of PN-afflicted skin. Eosinophils release granules containing neuropeptides, ECP, eosinophil-derived neurotoxin (EDN), eosinophil protein X (EPX), and major basic protein [12,34,35,36]. ECP and EDN are noteworthy due to their neurotoxic effect, and both are notably elevated in the skin of individuals with PN [34,35,37].

T cells and their cytokines, especially IL-31, play a role in the development of PN. The messenger RNA levels for the T cell-derived cytokine IL-31 are higher in the skin of individuals affected by PN compared to individuals with healthy skin [12,33,34,38]. IL-31 is primarily produced by activated Th2 cells, and the sensation of itching is propagated by IL-31 through its interaction with IL-31R (IL-31RA and oncostatin M receptor-β (OSMRβ) subunits) [33,34]. Transgenic mouse studies provide support for this mechanism, demonstrating a correlation between IL-31 expression and pronounced skin inflammation as well as intense itching [39]. Also, itch intensity has consistently shown a correlation with the abundance of dermal cells expressing IL-31 and oncostatin M (OSM) [40]. Conversely, the application of anti-IL-31 monoclonal antibodies resulted in a notable decrease in scratching reactions [34,41].

Elevated levels of Th2 cytokines, including IL-4, have been identified in skin lesions resembling prurigo [12]. Recent transcriptomic comparisons between affected and unaffected skin in PN emphasize the crucial involvement of IL-4 and IL-13 in disease pathogenesis. These investigations reveal distinctive alterations in fibroproliferative regulation, epithelium development, cornification, nervous system and vasculature development, and mesenchymal differentiation pathways specific to PN, distinguishing it from conditions like AD or psoriasis [42]. In PN, the mRNA expression of IL-4, IL-17, and IL-22 was notably elevated in affected skin [43], along with increased blood levels of IL-13 and periostin [6,44].

The Janus kinase (JAK) family includes four components (JAK1, JAK2, JAK3, and TYK2) which attach specifically in different arrangements to distinct type I/II cytokine receptors, transmitting activated internal transcriptional signals. This occurs alongside the involvement of seven members from the signal transducer and activator of transcription protein (STAT) family [45,46]. Key cytokines like IL-4, IL-13, IL-31, and OSM promote pruritus and inflammation in PN through the JAK-STAT signaling pathway, with a notable involvement of JAK1, STAT3, and STAT6 [45,47,48,49]. STAT6 predominantly regulates Th2 cytokines, while STAT3 is associated with various pruritogens, including IL-6, IL-22, IL-31, OSM, TSLP, and VEGF [45,50,51,52]. Lesional PN skin exhibits the significant upregulation of STAT3 and STAT6 [53,54,55], suggesting that JAK inhibitors may be effectively impede disease progression.

Patients with PN undergoing IL-4/IL-13 treatment exhibited a heightened expression of the activated STAT6 transcription factor in lesional skin [55]. The activation of STAT6 happens after IL-4R and IL-13R are triggered, with these receptors being prominently present not just in keratinocytes and immune cells, but also in fibroblasts and nerve cells [6,47]. IL-4 and IL-13 disrupt the neuroimmune network of the skin, leading to intense itching and PN [6,56]. These cytokines increase the expression of IL-31R and other receptors associated with itching, heightening the sensitization of histamine-independent sensory neurons to pruritogens [57]. In the context of PN, IL-4 and IL-13 also stimulate fibroblast proliferation, migration, and the production of pro-fibrotic factors such as TGFβ, collagen, periostin, and other extracellular matrix proteins. This process contributes to exaggerated tissue repair and the characteristic skin fibrosis seen in PN lesions [6,58].

The involvement of IgE in PN remains unclear, although comorbidities such as AD suggest shared pathogenic pathways including IgE. Some case reports indicate improved pruritus and skin lesions in PN following treatment with the anti-IgE medication omalizumab [6,59]. Mast cells play an increasingly recognized and distinct role in PN, serving as a major source of OSM and expressing high levels of IL-31RA in the skin of PN patients [40]. Numerous mast cell-associated mediators and receptors, including IL-4, IL-13, IL-31, Mas-related G protein-coupled receptor member X2 (MRGPRX2), and protease-activated receptor 2 (PAR2), have also been implicated in the pathogenesis of PN [6,40,60]. For instance, the severity of PN is correlated with the number of MRGPRX2-expressing cells, predominantly mast cells [6,61]. Interactions between mast cells and nerves are believed to contribute to itching in PN [47,62]. Intriguingly, the CXCL9/CXCL10/CXCR3 axis, known for its significance in neuroinflammatory diseases [63], has previously been implicated in itching and in this context may have a potential role in the pathogenesis of PN [64,65].

### 3.2. Neural Dysregulation

Previous studies have explored nerve fibers’ structure and arrangement in both affected and unaffected dermal areas of individuals with PN [34]. Pautrier documented the presence of neuronal hyperplasia in the dermis of a patient diagnosed with PN [33]. Various studies have confirmed this by using staining techniques for the pan-neuronal marker protein gene product (PGP)-9.5 and nerve growth factor (NGF) receptor in lesional PN skin [35]. One study verified that NGF receptor- and PGP-immunoreactive structures in the dermis of PN patients are significantly more dense than healthy controls. However, in the epidermis of PN patients, there is a lack of NGF receptor-immunoreactive nerve fibers, and fewer PGP-9.5+ nerve fibers [12,35]. Similarly, multiple studies have demonstrated a reduction in intraepidermal nerve fiber density [11,12,66,67,68], and following the resolution of itchy lesions, nerve fiber density returns to normal [68].

A recent study uncovered a distinct degree of branching in individual nerve fibers within the epidermis, which is potentially indicative of a reactive response; this was evaluated using a semi-quantitative pattern analysis. Similarly, there was an observed increase in the expression of genes associated with axonal growth, like NGF, in PN patients [66]. The altered neuroanatomy of the epidermis is linked to changes in nerve fiber function and neuronal hypersensitivity [66,69,70,71]. In chronic prurigo of diverse origins, hyperkinesis resulting from electrical stimulation has been detected, indicating common mechanisms of central neuronal sensitization [72]. In PN patients, stimulation with cowhage resulted in increased itch intensity in affected skin compared to healthy controls, indicating neuronal sensitization of mechano- and heat-sensitive C-fibers [70]. Another sign of neuronal sensitization is the heightened responsiveness to itching triggered by pinprick stimuli (punctate hyperkinesis), which is notably more pronounced at the location of itchy lesions compared to unaffected skin in PN patients. As Hashimoto et al. suggested, this could result from changes in the neural structures within the epidermis and the influence of immune processes [69]. Interestingly, pain-transmitting mechanisms remain unchanged in PN [72].

The components of the skin collaborate across the local nervous, immune, and endo-crine systems to uphold crucial functions of the epidermis. Within the epidermis, there exists a self-regulating neuroendocrine system, comprising hormonal and neuropeptide networks. These networks bolster the barrier function of the epidermis and fine-tune its responses to maintain optimal homeostasis, along with their corresponding receptors found in keratinocytes, melanocytes, Langerhans cells, and Merkel cells. Various neuropeptides are locally synthesized within the human epidermis, in addition to being released from nerve endings. Examples of these neuropeptides include the calcitonin gene-related peptide (CGRP), substance P (SP), and vasoactive intestinal peptide (VIP) [73,74,75,76,77]. In PN patients, scratching or neuronal activity leads to the activation of sensory nerves through both histaminergic and non-histaminergic pathways, initiating the release of these neuropeptides [62,78,79]. There is growing evidence pointing to the dysregulation of these neuropeptides, notably CGRP and SP, in the pathophysiology of PN (Figure 1). SP, a neurotransmitter released by nerve cells, interacts with two receptors, namely neurokinin-1 receptor (NK1R) and MRGPRX2 [12,32,33]. A study reported increased SP1 nerve fibers and a heightened SP expression in the dermal regions of PN-affected skin [33,34,80]. In individuals with PN, there is a significant increase in serum levels of SP and the upregulation of its receptor MRGPRX2 in the skin, when compared to healthy controls [61,79]. Another neuropeptide, CGRP, with a mechanism akin to SP, is elevated in PN and may be conducive to the pathogenesis of the disease by inducing neurogenic inflammation through eosinophils and mast cells [12,33,35]. CGRP can also influence endorphin levels, leading to the dysregulated expression of mu (MOR) and kappa (KOR) opioid receptors, potentially contributing to pruritus in PN [12]. Apart from mast cells, neuropeptides directly stimulate various immune cells in the skin, including macrophages, dendritic cells, T cells, innate lymphoid cells, and eosinophils. As mentioned earlier, these immune cells release inflammatory mediators such as histamine, tryptase, proteases, and pruritogenic Th2 cytokines like IL-4, IL-13, IL-31, TSLP, and IL-33 [62]. IL-31 plays a pivotal role in the neuroimmune connection between type 2 inflammation and sensory neurons [81]. IL-31R is expressed not only on skin sensory nerves but also on keratinocytes, mast cells, eosinophils, basophils, and monocytes. The expression of IL-31RA by sensory neurons is influenced by IL-4, which is associated with chronic severe itching. These inflammatory mediators heighten neuronal activation in inflamed skin, transmitting signals to nearby efferent neurons and the central nervous system, initiating an itch sensation that perpetuates the itch–scratch cycle [82]. Also, epidermal stress triggers itching through alarmins and other pruritogenic molecules (TSLP, IL-33, kallikrein, or cathepsin S) activating specific receptors on sensory neurons (TSLPR, ST2, MRGPRC11, and PAR2), thereby contributing to itching and sustaining the itch–scratch cycle [82].

Transient receptor potential (TRP) channels can respond to diverse physical and biochemical stimuli. Most itch receptors, like G-protein-coupled receptors and cytokine receptors, interact with TRP channels as downstream sensors [45,83]. When activated, these channels facilitate calcium influx and initiate action potentials that transmit itch signals through peripheral sensory neurons [45,84,85]. TRP channels constitute a group of nonselective cation channels, including TRP vanilloid 1 (TRPV1) and TRP ankyrin 1 (TRPA1) [45,86]. TRPV1, also known as the capsaicin receptor, is notably upregulated in nerve fibers and keratinocytes in affected skin areas of PN [45,87]. Itch signals, independent of histamine, involve TRPV1 and/or TRPA1. For instance, itching that is triggered by IL-31 and periostin is relayed through TRPV1+TRPA1+ neurons [45,88,89]. Itching that is mediated by IL-13, TSLP, endothelin, and MRGPRs is activated by TRPA1 [45,90,91,92,93]. Additionally, NGF activates TRPV1 via TrkA, leading to TRPV1 upregulation and activation in sensory nerves, releasing SP and CGRP afterwards [45,94]. More research is needed to identify the roles of TRPV1 and TRPA1 in itching related to PN [45].

### 3.3. Fibrotic Response

In PN, both type 2 and type 22 inflammatory responses are recognized as significant pathogenic features like in AD [95,96,97,98]. As previously described, elevated levels of type 2-associated mediators such as IL-13 and periostin are observed in the skin and peripheral blood in both conditions [44,99,100,101]. Interestingly, up to 50% of PN patients exhibit an atopic predisposition or active AD feature, leading some authors to speculate that PN might be a variant of AD rather than an independent disease entity [1]. However, recent data highlight unique functional characteristics in various cellular compartments of PN, allowing for a clear differentiation from AD. Two recent studies indicate that PN is distinguished by a robust fibrotic response in the stromal compartment, aberrant activation of keratinocytes, and a subdued type 2 inflammatory response in comparison to AD [102,103]. The stromal compartment exhibits activated fibroblasts and endothelial cells with heightened features of angiogenesis, fibrosis, and extracellular matrix production. PN appears to manifest a distinct mesenchymal dysregulation, setting it apart from AD, aligning with a well-established histological characteristic of PN—dermal fibrosis. At the epidermal level, keratinocytes in PN exhibit a markedly inflammatory phenotype and altered differentiation. Network and correlation analyses indicate that these changes result from a Th2/Th22 cell immune response, common to both AD and PN, possibly explaining the histological similarities such as hyperkeratosis and acanthosis. While the overall epidermal response does not reveal strong distinguishing features between AD and PN, a minor type 1 immune response is observed in PN keratinocytes, potentially reflecting cellular stress from repetitive scratching [104]. Concerning fibroblasts, a distinct subset characterized by CXCL14-IL24+ secretion in papillary fibroblasts, along with heightened levels of neuromedin B, has been discovered [102]. Another recent single-cell analysis has spotlighted fibroblasts, revealing an elevated presence of a cancer-associated fibroblast (CAF)-like phenotype, specifically those expressing WNT5A, in PN-afflicted skin lesions [105].

## 4. Clinical Features, Diagnosis, and Comorbidities

The quantity of nodules in PN can vary from a few to more than a hundred, typically arranged symmetrically on the extensor surfaces of the limbs and trunk. Most PN lesions measure between several millimeters up to 2 cm in diameter. Excoriation and crusting often accompany the nodules, which is indicative of an intractable itch–scratch cycle, with pruritus severe enough to cause bleeding in some cases [10]. While the skin between nodules is usually normal, it can be xerotic, thickened, or exhibit signs of postinflammatory pigmentary changes [36]. Pruritus is the main characteristic in PN, though some patients may experience burning or stinging pain [106].

Since PN is diagnosed clinically, comprehensive patient history taking and physical examination are important in assessing PN. Individuals with PN commonly report severe itching persisting for more than six weeks, which can be continuous, sporadic, or sudden [106]. It is crucial to gather a detailed medical history, including a review of medications and supplements, as well as an overview of all health conditions, including any psychiatric issues, which can assist in identifying PN-related conditions. During the physical examination, PN patients typically exhibit clusters of nodules on at least two distinct extensor areas, possibly extending to the trunk. While some PN patients may also present with concurrent skin conditions like AD, the presence of these hyperkeratotic and occasionally scratched nodules is indicative of PN [12].

The varied diagnoses that could be mistaken for PN encompass several conditions, including perforating disorders like Kyrle’s disease (though some specialists classify perforating lesions as a subset of PN), hypertrophic lichen planus, AD accompanied by lichen simplex chronicus, autoimmune blistering ailments such as bullous pemphigoid and dermatitis herpetiformis, neurotic excoriations, and body-focused repetitive behaviors associated with skin picking syndromes. Other potential differential diagnoses involve lichen amyloidosis, multiple keratoacanthomas, arthropod bites, and scabies [5]. Nodules of PN typically are firm to hard when touched, and their surface may appear thickened or have a sunken center. They often appear on the limbs, especially the extensor surfaces, with rare involvement of the face and palms. Although nodules can appear anywhere on the body, a typical pattern can be observed, known as the ‘butterfly sign’, which spares the skin on the upper back [107]. In patients with AD, the surrounding skin is often thickened and dry, with additional signs of atopy like a Dennie–Morgan fold or increased lines on the palms. Hypertrophic lichen planus typically manifests on the front of the lower legs and the joints between the fingers and toes. The primary lesions are raised, purple-red plaques and nodules with thickened skin and may develop wart-like growths. Follicular accentuation, elevation, and a chalky appearance are common characteristics of hypertrophic lichen planus lesions [108]. Though PN is primarily diagnosed clinically, obtaining a skin biopsy may be warranted for atypical presentations [12,106,109]. Characteristic histolopathologic features of PN display thick compact orthohyperkeratosis, pseudoepitheliomatous hyperplasia, focal parakeratosis, and hypergranulosis in the epidermis [5,109]. These features are clinically consistent with hard, hyperkeratotic, dome-shaped nodules that are dark brown in color [8]. Papillary dermal fibrosis with vertically arranged collagen fibers, increased numbers of capillaries and fibroblasts, and a mixed superficial, perivascular, or interstitial inflammatory infiltrate may be observed in the dermis [5,109]. These changes correlate with scratch-induced injury [12]. Direct immunofluorescence may be indicated to rule out autoimmune blistering disorders. Furthermore, skin scrapings may be helpful if scabies or fungal infections are suspected [5].

Recording the progression of PN is crucial for making informed therapeutic decisions [110]. Additionally, the standardized evaluation of itching across various conditions and medical specialties allows for comparisons of individual severity and impact, as well as variations based on geography and ethnicity. In 2008, the IFSI initiated efforts to standardize itch assessment tools globally. Currently, there is widespread agreement on the use of chronic prurigo assessment instruments, primarily relying on patient-reported outcomes (PROs) due to the absence of a singular biomarker [111]. Within the different domains of chronic prurigo, itch intensity serves as the primary PRO for evaluating the current severity and course of PN. Monodimensional scales prompt patients to rate itch intensity from no itch to the worst imaginable itch. The numeric rating scale (NRS) comprises 11 numbers (0: no itch; 10: worst itch imaginable), while the visual analog scale (VAS) features a 10 cm line, with anchors at both ends representing 0 and 10 (or 100) [112]. Both NRS and VAS are employed to assess the worst and/or average itch intensity experienced over the past 24 h. This assessment is commonly used as a primary endpoint in relevant trials, referred to as either worst itch NRS or peak pruritus NRS [113]. Multidimensional scales, incorporating intensity, scratching behavior, and sleep, are less frequently utilized due to challenges in unequivocal grade selection. The consequences of chronic prurigo can also be monitored and are typically employed as secondary endpoints [113].

Several coexisting conditions linked to PN have been revealed through case series and epidemiological investigations. Familiarity with these associated comorbidities can assist healthcare providers in conducting a thorough assessment and implementing an effective management plan for PN patients. Psychiatric disorders can either be provoked by the presence of PN or may exist alongside it as additional health conditions. Patients with PN often experience increased levels of depression, increased reliance on antidepressant medications, and a tendency towards suicidal thoughts [114]. Danish [114] and US [13] studies demonstrate increased odds of depression and anxiety in PN patients, often necessitating pharmaceutical intervention. The US study additionally indicates elevated rates of mood and anxiety disorders compared to individuals with other inflammatory skin disorders [13]. Additionally, a study involving 263 individuals diagnosed with chronic prurigo found that nearly all of them (97.2%) suffered from sleep disruptions due to itching [115]. It is understandable that disruptions in sleep and other psychological issues such as anxiety and depression play a substantial role in the considerable burden of illness faced by individuals with PN [116]. Understanding the interplay between mental health conditions and PN, whether contributing to disease development or exacerbated by chronic symptoms, warrants additional exploration [117].

PN is connected to various infectious agents, with HIV being extensively examined [118,119,120,121]. In HIV patients, PN is associated with severe itching and a notable decline in quality of life [118]. US and high HIV prevalence area (such as French Guyana) studies revealed an increased likelihood of PN in HIV-infected individuals [118,119]. Notably, PN lesions in HIV patients may respond to antiretroviral therapy [121,122]. Besides HIV, PN has been linked to other viral infections like hepatitis C, suggesting a potential association between persistent viral infections and PN pathogenesis [9,123,124,125].

Patients diagnosed with PN exhibit a high prevalence of celiac disease, Hashimoto thyroiditis, inflammatory bowel diseases, and type 1 diabetes mellitus [126,127,128]. A US epidemiologic study confirmed an increased rate of celiac disease in PN patients (OR 2.70, 95% CI 1.43–5.08) and identified associations with inflammatory bowel diseases, including Crohn’s disease (OR 2.40, 95% CI 1.51–3.81), ulcerative colitis (OR 1.64, 95% CI 1.13–2.37), and type 1 diabetes mellitus (OR 2.23, 95% CI 1.72–2.90) [13]. Meanwhile, PN commonly coexists with various dermatologic diseases, as the majority of PN patients also have another skin condition [25,129]. AD is frequently reported alongside PN, [16,27,130] and large-scale US studies indicate higher odds of allergic comorbidities, including asthma and urticaria, in PN patients compared to controls [13]. PN is associated with increased odds of psoriasis and neurotic excoriations [13]. Case reports and series additionally underscore connections with other skin diseases, such as keratoacanthomas, bullous pemphigoid, and linear immunoglobulin A disease [131,132,133,134].

PN is noted to have associations with cancer, particularly hematologic malignancies like non-Hodgkin’s and Hodgkin’s lymphoma [135,136,137]. PN can manifest as an initial symptom of lymphoma, and addressing the underlying lymphoma may lead to improvement or healing of PN lesions in some instances [135,136]. US-based epidemiologic studies suggest PN patients have two to five times greater odds of non-Hodgkin’s lymphoma compared to control subjects [13,138]. PN is also linked to primary cutaneous lymphoma, mycosis fungoides, multiple myeloma, and, in terms of solid tumors, potentially gastrointestinal tract cancers [138,139,140]. However, further research is necessary to validate these associations through extensive, multi-institutional studies.

Patients with PN frequently experience endocrine and metabolic dysfunction, impacting more than half of individuals in specific cohorts [130,141]. A significant association exists between PN and both type 1 and type 2 diabetes mellitus, with 2.23 (95% CI 1.72–2.90) and 1.42 (95% CI 1.30–1.55) times increased odds, respectively, compared to matched control subjects [13]. PN patients also demonstrate elevated odds of hypertension, hyperlipidemia, and obesity in comparison to the general population [13,16]. The hypothesis suggests that pruritus resulting from underlying metabolic dysregulation may contribute to the onset of PN, although this mechanism is not fully comprehended [9,141].

In addition, PN is associated with systemic diseases affecting diverse organ systems, including renal, pulmonary, and cardiovascular systems. PN consistently correlates with kidney dysfunction, especially end-stage renal disease, presenting increased odds of chronic kidney disease (CKD) and dialysis [142,143,144]. The elevated pruritus experienced by individuals with CKD may heighten the PN risk in this group [145,146]. Chronic obstructive pulmonary disease may be more prevalent among PN patients, emphasizing the need to investigate tobacco use as a modifiable risk factor population [13,16]. Also, patients with PN exhibit increased risks of cardiovascular and cerebrovascular diseases. According to an epidemiologic study, PN patients have roughly double the odds of heart failure, cerebrovascular disease, and coronary heart disease compared to age- and sex-matched control subjects [13]. Notably, the odds of cerebrovascular and coronary heart disease in PN patients are higher even when compared to individuals with psoriasis, a condition already known for increased risks of atherosclerotic conditions and mortality [147]. The substantial comorbidity burden in PN, surpassing that of other inflammatory skin disorders like psoriasis, emphasizes the necessity of epidemiologic evaluation for mortality in PN patients.

As PN is associated with various comorbid conditions, it is crucial to detect systemic diseases conducive to PN, particularly in patients without any dermatoses [12,106,148]. A targeted laboratory work-up is recommended, encompassing a complete blood cell count with differential liver and renal function tests, and, depending on risk factors and clinical evaluation, thyroid function test, hemoglobin A1c/diabetes screening, HIV serology, and hepatitis B and C serologies [12,106,149]. Additional tests, including serum immunofixation, serum and urine protein electrophoresis, urinalysis, chest radiograph, stool examination for ova and parasites, and iron studies, should be considered based on the clinical history and review of systems [12,106,148]. PN patients should also undergo age-appropriate malignancy screening, with increased apprehension for associated malignancy in those with acute rather than chronic pruritus onset (<1 year) [12,106,109].

## 5. Treatment and Potential Therapeutics

The primary objective in managing PN is to disrupt the itch–scratch cycle and alleviate itching to promote the healing of nodules [10]. The current therapies for PN encompass various options such as mild skincare, soothing moisturizers to relieve itching, creams containing corticosteroids or calcineurin inhibitors, capsaicin-based topicals, medications with a neuromodulating effect, antidepressants, phototherapy, and drugs that suppress the immune system [5,150]. Decisions regarding PN treatment should rely on clinical assessment rather than adhering strictly to a predetermined sequence of steps [5]. Eventually, effective treatment of PN requires addressing both its neural and immunological aspects. A personalized treatment approach, tailored to factors like the age of the patient, co-existing health conditions, disease severity, and potential treatment side effects, is essential. Often, the most effective strategy involves a combination of systemic and topical treatments [27].

### 5.1. Topical Therapy

Various topical therapies for PN have undergone evaluation in randomized clinical trials, including corticosteroids, pimecrolimus, and calcipotriol. However, the limited effectiveness of these treatments underscores the ongoing need for research to develop more targeted and efficacious topical therapies for PN [12]. High-potency topical corticosteroids, such as betamethasone valerate 0.1% tape, remain the first-line topical therapy for PN. Studies have demonstrated that this tape can reduce pruritus and flatten nodules in PN patients compared to moisturizing itch relief cream alone [151]. Flurandrenolide tape, acting as a medicated occlusive skin barrier, has also shown effectiveness when applied to specific nodules, sparing surrounding skin. Occlusive treatments not only enhance the effects of the medication but also act as a physical deterrent against scratching [12].

Thicker PN lesions may necessitate the direct injection of corticosteroids into the affected areas, sometimes accompanied by cryotherapy [142,152]. A recent review suggested that a corticosteroid solution with a concentration of 2.5 mg/mL was both safe and efficacious for patients with localized dermatitis, including PN [152]. A study advised restricting the use of intralesional corticosteroids to patients with fewer than ten lesions to minimize side effects [5].

Topical anesthetics, including over the counter 1% pramoxine lotion, lidocaine spray, and compounded topical anesthetic creams, offer an alternative antipruritic option, providing modest itch relief for mild PN [12].

Nonsteroidal topical agents, including pimecrolimus, vitamin D derivatives, and capsaicin, have been investigated for PN treatment. A study demonstrated the efficacy of 1% pimecrolimus cream in reducing itching in PN patients [153]. Using topical capsaicin effectively relieves itching in PN patients by restoring normal TRPV1 expression in skin lesions and reducing levels of SP and CGRP [154]. The anti-itch effect of capsaicin is likely due to TRPV1 activation on cutaneous C-fibers, resulting in the reduction of neuropeptides such as SP [155]. However, due to short duration of relief of capsaicin and its side effect of severe burning sensations, its extensive utilization is restricted [12,45]. Topical TRPV1 antagonists (Figure 2A), tested in phase 2b (PAC-14028) and phase 3 (Asivatrep) clinical trials for treating atopic itching, have shown both efficacy and safety [156,157]. Yet, their effectiveness in treating PN needs confirmation through randomized controlled trials (RCTs) [45].

### 5.2. Systemic Therapy

Patients with PN often need systemic therapies as many are unresponsive to the previously mentioned treatments. Systemic options for PN treatment encompass phototherapy, immunosuppressants, gabapentinoids, antidepressants, and MOR antagonists [27,150]. PN, being a nonhistaminergic itch condition, generally does not respond well to antihistaminergic agents, except for their sedative properties, and their use is not suggested unless a comorbid histamine-mediated condition is suspected [12,27,107]. Considering the crucial role of immune dysregulation in PN pathogenesis, systemic immunosuppressants are commonly used [12].

#### 5.2.1. Phototherapy

Ultraviolet light therapy, known for its anti-inflammatory effects, proves effective in alleviating pruritus in various skin conditions, including PN [158]. This option is particularly valuable for medically complex patients with limited treatment choices due to comorbidities and potential drug interactions [27]. Among ultraviolet light therapies, narrowband ultraviolet B light therapy administered 2 to 3 times a week is considered the primary treatment for PN patients. Other modalities such as ultraviolet A light and ultraviolet B light have demonstrated some efficacy in PN treatment [12,27]. While phototherapy combined with topical therapy may be adequate for a minority of PN patients, the majority will likely require supplemental systemic therapy.

#### 5.2.2. Systemic Immunomodulating Agents

Immunosuppressants

Retrospective studies provide evidence for the effectiveness of immunosuppressants like methotrexate and cyclosporine in treating PN [16]. Two retrospective studies focusing on methotrexate as a PN treatment demonstrated significant relief from pruritus and healing of lesions [159,160]. Due to its favorable side effect profile, methotrexate is frequently used as frontline immunosuppressive therapy, typically starting at around 15 to 20 mg weekly alongside topical therapy at the Johns Hopkins Itch Clinic [12]. In more severe cases, cyclosporine may be considered. Reports indicate that cyclosporine at a dosage of 2 to 5 mg/kg/day resulted in symptom improvement and nodule resolution [161,162]. However, a notable drawback to cyclosporine treatment is its significant side effect, necessitating regular monitoring of blood pressure, renal function, hepatic function, and complete blood cell counts [12,161,162].

Limited evidence suggests that azathioprine may provide a temporary reduction in pruritus for individuals with PN; however, the effects seem to be short-term. The use of azathioprine is associated with notable adverse events, including nausea, diarrhea, and epigastric pain, and vigilant monitoring for potential bone marrow suppression is advised [163]. In a retrospective analysis of 96 patients treated with azathioprine for pruritus, about 65% experienced suspected drug-related adverse events, leading to a permanent discontinuation of treatment in 33% of cases [164]. The prolonged use of systemic corticosteroids, such as prednisone, poses the risk of systemic complications due to the typical duration and dosage required, thus limiting their suitability for PN. The utilization of prednisone should be restricted to a short trial period, preferably lasting less than four weeks.

2.Targeting Th2 polarization (Dupilumab, Tralokinumab)

Dupilumab, an IL-4Ra monoclonal antibody (Figure 3A) that simultaneously blocks IL-4 and IL-13, has been explored for PN treatment, with initial case reports demonstrating encouraging outcomes [165,166,167,168,169]. Following the completion of two phase 3 clinical trials, dupilumab has gained approval from the US Food and Drug Administration (FDA) [165], European Medicines Agency (EMA) [6], and China National Medical Products Administration (NMPA) [45] as the first treatment option for adult patients with PN. These trials evidenced its effectiveness in alleviating both pruritus and PN lesions. The studies revealed response rates of approximately 58% and 60% for achieving a ≥4-point improvement on the worst itch numerical rating scale (WI-NRS), which ranges from 0 to 10. Moreover, rates of skin lesion resolution, complete or nearly complete, were observed in up to 48% of patients after undergoing 24 weeks of treatment [170]. Furthermore, anecdotal evidence suggests that off-label use of dupilumab has demonstrated favorable efficacy and safety profiles in pediatric and adolescent PN patients [171,172,173]. However, dupilumab took longer to alleviate pruritus in atopic PN compared to nonatopic PN patients [174]. Real-world studies have evaluated the long-term effectiveness of dupilumab, with a significant proportion of PN patients reporting improved pruritus at weeks 16 and 52 [175], and some experiencing sustained relief for over 104 weeks [176]. A systematic review revealed that nearly half of PN patients achieved complete relief from itching, with an average time to symptom resolution of 19 weeks, although those who did not achieve complete resolution experienced delayed relief [174]. Despite these findings, there remains a need for extensive real-world data to comprehensively assess the efficacy and safety of dupilumab in PN. Additionally, novel drugs targeting Th2 polarization in PN have demonstrated encouraging outcomes, such as tralokinumab, an anti-IL-13 monoclonal antibody, which demonstrated a significant reduction in itch severity as early as week 4 in a case series of patients with a PN-like phenotype of AD [45,177].

3.JAK inhibitors

JAK inhibitors (JAKi) are a class of small molecule drugs that inhibit various signal transduction pathways (Figure 3B) [50,178]. While biologics typically achieve their maximum efficacy after 16 weeks or more [178], systemic JAK inhibitors act quickly [179], providing a rapid relief (within days) of key symptoms such as itching and skin pain [178,180,181]. JAK inhibitors interact with multiple cytokine receptors that utilize JAK1 and/or JAK2 in their signaling, offering a broader spectrum of action compared to more targeted biologics. However, this broader action also reflects their safety profile. While current biologics generally have minimal adverse effects, JAK inhibitors are associated with potential adverse reactions including virus-induced infections (such as zoster or eczema herpeticum), major cardiac events, or venous thromboembolism, particularly in patients at risk [182,183,184,185,186].

Several case reports have demonstrated the effective treatment of refractory PN using tofacitinib (a JAK1/3 inhibitor), baricitinib (a JAK1/2 inhibitor), and upadacitinib (a JAK1 inhibitor) [45,187,188,189,190,191,192,193,194,195,196]. Ongoing developments include phase 2 clinical trials investigating the efficacy of two JAK1 inhibitors, abrocitinib (NCT05038982) and povorcitinib (NCT05061693), for PN treatment, as well as a phase 3 clinical trial assessing the use of ruxolitinib cream (a JAK1/JAK2 inhibitor) (NCT05755438 and NCT05764161).

Although the overall rate of side effects reported in JAK inhibitor phase 3 trials was low, their safety profiles underwent rigorous regulatory evaluation by national competent agencies such as the FDA in the United States and the EMA in Europe. Abrocitinib, baricitinib, and upadacitinib have demonstrated significant effectiveness in reducing itching in patients with AD [197]. Since these medications target JAK1 and/or JAK2, probably along with other JAK inhibitors, they will exhibit efficacy in PN as well [178,179]. Recent case reports have highlighted the effectiveness of baricitinib in PN [192,193,194], and ongoing phase 2 trials are assessing the efficacy of the JAK1 inhibitors abrocitinib and povorcitinib in treating PN. Further long-term treatment studies are necessary to evaluate the safety and tolerance of these medications in older patients with PN, who often have more accompanying health conditions compared to those with AD.

#### 5.2.3. Systemic Neuromodulating Agents

Antidepressant

Antidepressants like paroxetine, fluvoxamine, duloxetine, and amitriptyline offer mild to moderate relief from pruritus [12,198,199,200,201]. For example, paroxetine (initiated at 10 mg daily for 3 days, followed by maintenance dosing at 20–60 mg daily) or fluvoxamine (commencing at 25 mg daily for 3 days, followed by maintenance dosing at 50–150 mg daily) have demonstrated efficacy in reducing itching in PN patients. Duloxetine, an antidepressant approved for neuropathic pain, at a daily dose of 20 to 60 mg, may also contribute to alleviating itching associated with PN [12,199]. In patients with depressive disorder and PN, it could be a promising alternative treatment. Lastly, a pilot study revealed positive responses in several PN patients treated with amitriptyline, starting at 60 mg daily for 3 weeks, followed by 30 mg daily for 2 weeks, and 10 mg daily for 1 week.

2.Gabapentinoids

The treatment for PN includes various agents focusing the neural pathogenesis of itch transmission, spanning from neural innervation in the skin to the dorsal root ganglion and progressing through the spinal cord to the brain [12]. Gabapentinoids, such as gabapentin and pregabalin, are commonly prescribed to reduce itching by inhibiting calcium signaling [202]. Elderly patients are typically initiated at low doses (e.g., 100 mg nightly) and gradually titrated upwards due to the risk of significant sedation. In contrast, younger patients may begin at 300 mg nightly and be titrated upwards to a maximum of <3600 mg daily, divided into thrice-daily dosing. Pregabalin, with a similar mechanism of action, is gradually titrated upwards with doses ranging from 75 to 600 mg daily. While these agents may be effective in specific patient subsets, the need for higher doses to alleviate intense itching can lead to significant sedation, which is a common reason for discontinuation of treatment [12].

3.Neurokinin-1 receptor antagonists (aprepitant, serlopitant)

A different class of drugs that has demonstrated some effectiveness in targeting the neural origins of itching includes NK-1R antagonists, which are thought to alleviate itching by blocking SP (Figure 2B) [203,204]. One such agent in this category, aprepitant, has FDA approval for chemotherapy-induced nausea and vomiting. In its off-label use for chronic pruritus, the prescribed dose can vary based on the underlying disease, with a common prescription being 80 mg daily [12,203,205]. An open-label study suggested that aprepitant might be effective in reducing itching associated with PN, but a randomized phase 2 trial did not demonstrate efficacy in reducing itch severity in PN [12,205]. Similarly, serlopitant, another neurokinin-1 receptor antagonist, showed notable improvement in itching compared to placebo as early as two weeks after starting treatment in a phase 2 trial [204]. However, these findings were not replicated in the phase 3 trial, which failed to meet its primary endpoint. One potential explanation is that SP may also signal through MRGPRX2, which is not affected by blocking NK-1R. Preclinical studies are exploring MRGPRX2 antagonists as a potential solution [206].

4.Thalidomide

Thalidomide is a neuroactive medication considered for patients with PN resistance to standard treatments, typically administered at doses ranging from 50 to 150 mg daily [207,208,209]. However, its use demands extreme caution due to recognized neurotoxic and teratogenic effects, leading to an elevated risk of peripheral neuropathy and birth defects, especially in pregnant women [207]. Thalidomide should be reserved for individuals who have not experienced improvement with conventional therapeutic approaches.

5.Targeting opioid receptor (Nalbuphine, Butorphanol, Naltrexone)

At the spinal cord level, an imbalance in the activity of the itch-promoting MOR and the itch-inhibiting KOR is implicated in non-histaminergic itching [210]. KOR agonist/MOR antagonist drugs like nalbuphine and butorphanol (Figure 2C) have demonstrated potential benefits [211,212,213,214]. Findings from a phase 2 placebo-controlled study demonstrated the effective treatment of PN with the dual-acting KOR agonist/MOR antagonist nalbuphine extended-release (ER) tablets. Notably, 33% of participants receiving 162 mg oral doses twice daily experienced a reduction of ≥50% in itching at week 10, demonstrating a favorable safety profile [45,215]. The phase 2b/3 PRISM clinical trial assessing nalbuphine ER met its primary endpoint, revealing a higher proportion of participants achieving a ≥4-point reduction in WI-NRS at week 14 compared to those on placebo (24.7% vs. 13.9%) [45,211,212,213,214,215,216]. Intranasal butorphanol, specifically 1 mg as needed, has been employed in PN cases, including those resistant to standard treatments, aiming to disrupt the itch–scratch cycle [211]. Additionally, the MOR antagonist naltrexone (Figure 2C) (50 mg) has exhibited anti-pruritic effects in certain subsets of PN patients [212].

6.Cannabinoids

Regarding cannabinoids, cutaneous nerve fibers expressing cannabinoid receptors 1 (CB1) and 2 (CB2) are believed to play a role in itch sensations. A systematic review has indicated a substantial relief of symptoms in patients with chronic pruritus who received cannabinoid treatment after not responding to initial therapies [217]. CB1 and CB2 are found in both the central nervous system and skin nerve fibers, responding to diverse bioactive lipid mediators [218]. Various studies in animals with chronic pruritus and clinical trials on pruritic skin conditions demonstrate the effectiveness of cannabinoid agonists in relieving itching [216,219,220,221,222,223]. In an open-label clinical study, a moisturizing cream containing the CB2 agonist N-palmitoyl ethanolamide (Figure 2D) significantly alleviated itching in eight out of twelve patients with PN [45,224]. However, further confirmation of the involvement of the endogenous cannabinoid system in PN is needed through double-blind controlled trials.

### 5.3. Up-and-Coming Therapeutic Agents

Targeting IL-31 (Nemolizumab)

In response to the heightened IL-31 levels observed in PN patients, targeting IL-31 has emerged as a promising therapeutic approach. IL-31 interacts with a heterodimeric receptor composed of IL-31RA and OSMRβ, influencing neuroimmune signaling in PN and other pruritic conditions like AD [40,225]. Nemolizumab acts by inhibiting IL-31RA (Figure 3C) and serves as an additional therapy for AD in patients aged ≥13 years in Japan, particularly when prior treatments have proven insufficient [225]. A study led by Ruzicka et al. investigated the effects of subcutaneous nemolizumab, observing a significant alleviation of itchiness in individuals with AD [226]. Nemolizumab stands as the sole systemic therapy under investigation in three phase 3 RCTs (NCT05052983, NCT04204616, NCT04501666; refer to ClinicalTrials.gov, EudraCT clinicaltrialsregister.eu) [227]. Results from the OLYMPIA 2 phase 3 RCT (NCT04501679) involved 274 moderate-to-severe PN patients globally. Among them, 183 received a bodyweight-dependent nemolizumab dosage after a 60 mg loading dose (<90 kg: 30 mg every 4 weeks; ≥90 kg: 60 mg every 4 weeks) for 16 weeks, while 91 received a placebo. At the 16-week mark, both primary endpoints, i.e., the proportion of patients with a pruritus improvement of ≥4 points measured using the weekly average Pruritus Numerical Rating Scale (PP-NRS) and the proportion of patients with Investigator’s Global Assessment (IGA) success, were met (*p* < 0.0001). A pruritus reduction of ≥4 points was observed in 56.3% of nemolizumab-treated patients (weekly average PP-NRS) compared to 20.9% in the placebo group. IGA success was noted in 37.7% of the nemolizumab group versus 11.0% in the placebo group. The most common individual adverse events were headache (6.6% vs. 4.4%) and AD (5.5% vs. 0%) [228]. In a 12-week, double-blind, phase 2 RCT (NCT03181503), rapid improvements in pruritus intensity (PP-NRS −19.5% vs. −5.8% [placebo]; *p* = 0.014) and sleep were observed within 48 h in nemolizumab-treated patients [229,230]. After 4 weeks, PP-NRS was reduced by 4.5 points (nemolizumab) compared to 1.7 points (placebo group; *p* < 0.001). At this point, almost 30% of nemolizumab-treated patients reported (almost) no pruritus, and 24% achieved ≥75% healed skin lesions (versus 11% in the placebo group). Adverse events were reported in approximately 70% of patients in both groups (severe events: n = 4 [nemolizumab]; n = 3 [placebo]). Gastrointestinal symptoms (21% of patients), musculoskeletal or connective-tissue symptoms (18% of patients), and bronchitis (6% of patients) were the most common adverse events in the nemolizumab group [229]. Post hoc analyses demonstrated a decrease in itching within just 2 days after treatment [230].

2.Targeting Oncostatin M beta receptor (Vixarelimab)

The OSMRβ, a recent therapeutic target, addresses lymphocytes to induce collagen production in dermal fibroblasts. Vixarelimab (KPL-716) a monoclonal antibody targeting OSMRβ, the shared receptor for IL-31 and OSM (Figure 3D), disrupts IL-31 signaling and is currently under assessment in AD and PN [231]. In two phase 2 RCTs (NCT03816891, NCT03858634), vixarelimab demonstrated a rapid reduction in pruritus intensity and substantial improvement of skin lesions. Sofen et al. conducted a study involving 50 patients in Canada and the US (NCT03816891), where participants were treated with vixarelimab (720 mg subcutaneous loading dose, followed by 360 mg subcutaneous weekly) or a placebo for 8 weeks. In the vixarelimab group, a significant reduction in pruritus, measured by the average weekly PP-NRS, was observed at week 8 (−50.6%) compared to baseline (placebo group: −29.4%; 95% confidence interval for the difference −40.8 to −1.6; *p* = 0.03). At this juncture, 52.2% of patients in the vixarelimab group achieved a pruritus reduction of ≥4 points (PP-NRS) compared to 30.8% in the placebo group (*p* = 0.11). Additionally, the IGA stage was 0 or 1 in 30.4% of patients in the vixarelimab group (versus 7.7% in the placebo group; *p* < 0.03) at week 8. Adverse events in the vixarelimab arm included upper respiratory tract infections (21.7% of patients), nasopharyngitis (13.0% of patients), nummular eczema, and urticaria (each 8.7% of patients) [232].

3.Targeting IL-5 (Benralizumab)

IL-5 plays a crucial role in eosinophil biology, contributing to differentiation, proliferation, activation, adhesion, and survival. Moreover, it facilitates the recruitment of mast cells and basophils to the skin [84,233]. Benralizumab targets the IL-5Rα subunit (Figure 3E), leading to eosinophil depletion, and it has gained approval for treating severe eosinophilic asthma [229,230]. Benralizumab is currently undergoing examination in a phase 2 trial for PN [231,232].

4.Targeting KIT (Barzolvolimab)

Inhibiting mast cell activation results in the suppression of the neuro-immune axis [233,234]. Barzolvolimab (CDX-0159), a monoclonal antibody (mAb) targeting mast cell tyrosine kinase KIT receptors (Figure 3F), has demonstrated the sustained and significant inhibition of mast cells in healthy volunteers. KIT interacts with stem cell factor (SCF) and is crucial for mast cell proliferation, migration, and survival [235]. A phase 1 clinical trial investigating the use of this drug for PN has concluded, and the findings are pending (NCT04944862).

## 6. Conclusions

In this article, we have focused on advancing our understanding of the origin of PN and identifying novel targets for effective therapies to address this persistent and challenging condition. Approaches centered on halting the transmission of itch signals or interfering with communication between the nervous and immune systems are emerging as potential therapies to relieve itching, prevent chronicity, and improve the overall outlook for PN.

The field of PN in dermatology has grappled with misconceptions and terminological confusion, impeding scientific progress for decades. Subsequent milestones include advancements in disease understanding, definition of terminology, formulation of guidelines, and the approval of the first therapy, dupilumab, in 2022. With innovative treatments on the horizon for PN, it is crucial for healthcare providers to enhance their understanding of the pathogenesis and current management of PN. Recent research has presented compelling evidence in elucidating the precise mechanisms that contribute to itching in PN patients. It is essential to delve deeper into distinct pruritogens and receptors independently and explore the intricate network interactions between skin cells, the nervous system, and fibroblasts to achieve a thorough comprehension of PN.

However, several studies are ongoing, including understanding the natural course of the disease and differences in molecular and treatment responses among different ethnicities. Further research, including international collaborations, registries, and translational studies, is deemed necessary and is already in progress.

## Figures and Tables

**Figure 1 ijms-25-05164-f001:**
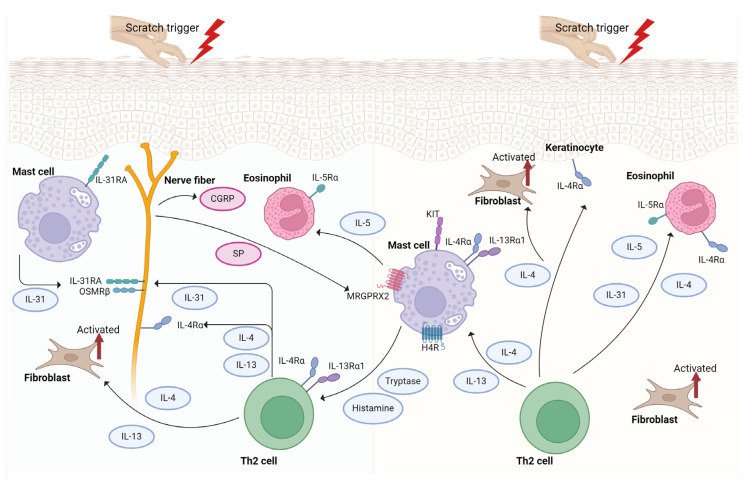
Overview of the mechanism of prurigo nodularis (PN). Crosstalk between keratinocytes, immune and inflammatory cells, and nerve fibers plays a pivotal role in PN. Keratinocytes serve as a significant source of growth factors and inflammatory cytokines, triggering immune activation. This results in increased infiltration of immune cells such as Th2, eosinophils, and mast cells, initiating inflammation and promoting keratinocyte hyperproliferation. Concurrently, dermal neuronal hyperplasia releases neuropeptides like substance P, further activating the immune response by interacting with immune cells and keratinocytes. The itch–scratch cycle exacerbates inflammation, while scratching causes mechanical damage to peripheral nerve fibers in the epidermis. SP, substance P; CGRP, calcitonin gene-related peptide; Th2 cell, T-helper 2 cell. This figure was created with Biorender at www.biorender.com (accessed on 11 March 2024).

**Figure 2 ijms-25-05164-f002:**
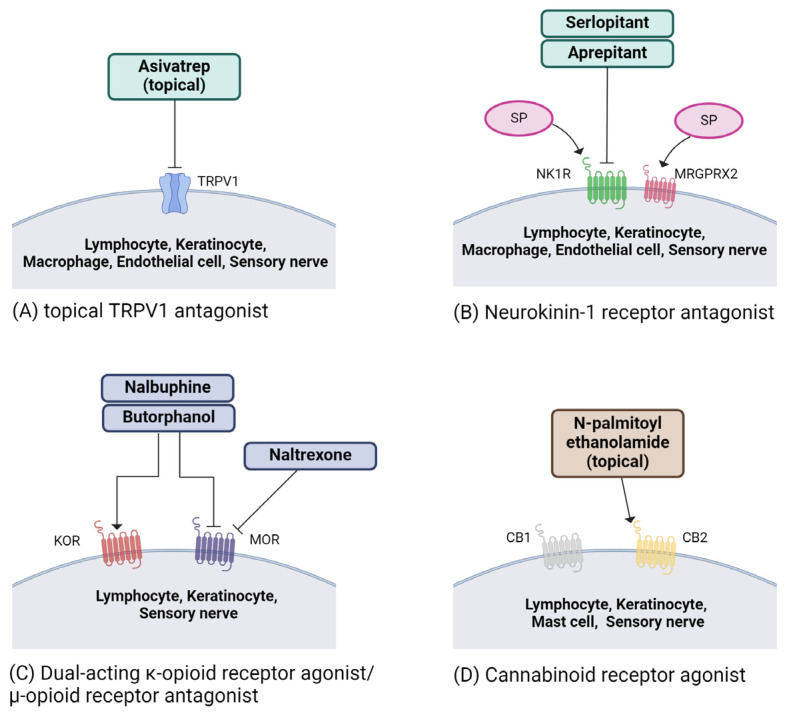
Neuromodulating treatments of current and promising therapeutic targets in prurigo nodularis (PN). (**A**) When activated, transient receptor potential (TRP) channels initiate itch signals through peripheral sensory neurons. TRPV1, also known as the capsaicin receptor, is notably upregulated in nerve fibers and keratinocytes in affected skin areas of PN patients. Topical asivatrep inhibits TRPV1. (**B**) Skin sensory nerves express numerous receptors associated with itching and interact with various inflammatory cells. This interaction leads to the release of neuropeptides such as substance P (SP), and it activates two receptors, neurokinin-1 receptor (NK1R) and Mas-related G-protein-coupled receptor X2 (MRGPRX2). Serlopitant and aprepitant blocks NK-1R. (**C**) Disparity of activity between the itch-promoting mu opioid receptors (MOR) and the itch-inhibiting kappa opioid receptors (KOR) is suggested to contribute to non-histaminergic itch. Drugs such as nalbuphine and butorphanol act as both KOR agonists and MOR antagonists, while naltrexone acts as a MOR antagonist. (**D**) Both CB1 and CB2 receptors are distributed in the central nervous system and skin nerve fibers. CB2 receptors are present in various peripheral immune cells. CB2 binding leads to a reduction in inflammation, particularly evident in dermatitis models. N-palmitoyl ethanolamide is a CB2 agonist. NK-1R, neurokinin 1 receptor; OR, opioid receptor; CB, cannabinoid receptor; TRP, transient receptor potential; TRPV1, TRP vanilloid. This figure was created with Biorender at www.biorender.com (accessed on 9 April 2024).

**Figure 3 ijms-25-05164-f003:**
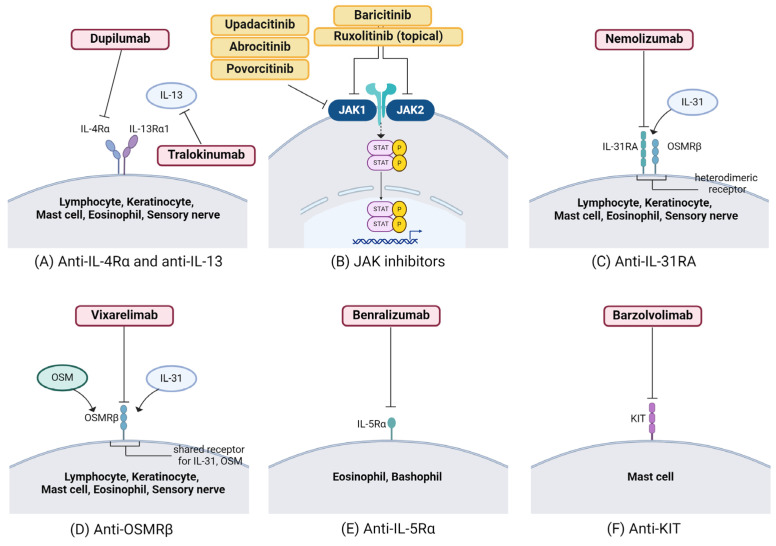
Immunomodulating treatments of current and promising therapeutic targets in prurigo nodularis (PN). Molecules are expressed by specific cell types and targeted by monoclonal antibodies and small molecules for the treatment of PN. Skin immune cells, keratinocytes, and fibroblasts release a variety of itching mediators independent of histamine. These mediators directly or indirectly stimulate receptors on nerve endings in the skin, leading to the release of neuropeptides, thus establishing a positive feedback cycle. Disrupting this cycle represents a pivotal strategy for innovative therapeutic approaches. These include (**A**) dupilumab targeting IL-4Rα, tralokinumab targeting IL-13, (**B**) Janus kinase inhibitors (JAKi), (**C**) nemolizumab targeting IL-31RA, (**D**) vixarelimab targeting oncostatin M receptor-β (OSMRβ), (**E**) benralizumab targeting IL-5Rα, and (**F**) mast cell-depleting therapy with barzolvolimab targeting KIT. Key molecules include IL, interleukin; JAK-STAT, janus kinase-signal transducer; OSM, oncostatin M. This figure was created with Biorender at www.biorender.com (accessed on 11 April 2024).

## Data Availability

The data presented in this study are available in the article.

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
