# Peer review of "Prurigo Nodularis: Pathogenesis and the Horizon of Potential Therapeutics"

_ijms, 2024, doi:10.3390/ijms25105164_

Round 1

Reviewer 1 Report

Comments and Suggestions for Authors

It is an excellent paper about prurigo nodularis that cover very well the etiopathogenetic and the therapeutical options. Hereby please find my comments regarding the paper:

1.      Please insert a more detailed descriptions that differentiate PN from other skin diseases.

2.      It is necessary to insert a discussion about the histopathology of PN and the correlation between clinical and histopathological aspects.

3.      In treatment section please insert a discussion on general aspects and modification of the lifestyle of these patients.

4.      The conclusions are well presented.

5.      The references are appropriate.

6.      The quality of figures and the data is good and the explanation of the figures is well done.

Recommendation: minor revision.

Reviewer 2 Report

Comments and Suggestions for Authors

The manuscript is interesting and well-written. The Authors presented in great detail the pathogenesis and treatment options of prurigo nodularis.

I recommend the manuscript for publication after making the following revisions.

Comments:

1.       The Authors use two abbreviations for US: US and U.S.

2.       I found the following punctuation errors:

·         Koreanepidemiological

·         compriseelevated

·         periostin [6] [44].

·         family. [45, 46].

·         [6] [47].

·         PN [6] [56].

·         Currently, Benralizumab is

3.       In the text of the manuscript there is no reference to a literature item 42.

4.       Figures 1 and 2 do not show the following descriptions appearing in the figure caption:

·         This results in increased infiltration of immune cells such as Th2, Th17/IL-17, Th22/IL-22, eosinophils, and mast cells, initiating inflammation and promoting keratinocyte hyperproliferation.

·         This interaction involves molecules like IL-31/IL-31R and 470 Mas-related G-protein-coupled receptor X2 (MRGPRX2), leading to the release of neuropeptides 471 such as substance P (SP).

5.       Figure 2 does not show exactly how neuromodulators affect PN symptoms relief.

6.       In my opinion, the following locations seem to be missing references:

·         line 313

·         line 445

·         line 600

7.       In the following passages, the same reference is unnecessarily given multiple times:

·         lines 280-287

·         lines 669-687

·         lines 691-697

·         lines 707-716

8.       The Authors unnecessarily indicate a reference to item 221, even though they do not cite it. The same is most likely true of item 225.

9.       In my opinion, the Authors can remove the list of abbreviations because they did not include all the abbreviations used.
